# Treatment of Unstable Pediatric Tibial Shaft Fractures with Titanium Elastic Nails

**DOI:** 10.3390/medicina55060266

**Published:** 2019-06-10

**Authors:** Abuzer Uludağ, Hacı Bayram Tosun

**Affiliations:** Department of Orthopaedics and Traumatology, Faculty of Medicine, Adiyaman University, 02100 Adiyaman, Turkey; bayramtosun@hotmail.com

**Keywords:** pediatric, fracture, tibia, elastic intramedullary nail

## Abstract

*Background and objectives*: Pediatric tibial shaft fractures often have satisfactory outcomes after closed reduction and casting. However, surgical treatment may be required in unstable or open fractures. Titanium elastic nails (TENs) are a good option for the surgical treatment of pediatric tibial fractures due to their advantages such as short hospitalization periods, easy applicability, early weight bearing, and early union. In this study, we evaluated radiological and functional outcomes in pediatric patients with tibial shaft fractures that underwent fixation with TENs. *Materials and methods*: A total of twenty tibial shaft fractures that were treated with TENs in our clinic between 2013 and 2017 were retrospectively reviewed. The mean age at injury was 8.9 ± 2.78 (range of 3–14) years. Seven (35%) out of 20 fractures were open fractures, of which one fracture was classified as Grade I and six fractures were classified as Grade II. In each patient, antegrade nailing was performed by inserting a TEN in the medial and another TEN in the lateral side of the proximal metaphysis. Clinical outcomes including union, alignment, leg-length inequality, and complications were evaluated using modified Flynn’s criteria. *Results*: The mean time to union was 10.85 ± 3.39 (range of 6–20) weeks. No patient had a sagittal or coronal angulation of over 10°. One patient had a leg-length inequality of 10 mm. Among three patients with open fractures, two of them had superficial wound infections and the other patient had a deep wound infection. All the infections were successfully treated with appropriate antibiotic therapies. Four other patients had pin tract irritation that required no intervention. No significant difference was observed between patients with open and closed fractures with regard to the clinical and radiological findings although patients with open fractures had a significantly higher complication rate compared to patients with closed fractures (*p* < 0.05). No patient had a restricted range of motion of the ankle and knee joints. Twelve (60%) patients had an excellent outcome, and eight (40%) patients had a satisfactory outcome. *Conclusions*: Intramedullary fixation with TENs provides favorable outcomes and reduced complication rates in the treatment of unstable pediatric tibial shaft fractures that cannot be reduced with conservative treatment modalities or cannot be casted due to the presence of an edema or open wound.

## 1. Introduction

Tibial shaft fractures are the third most common long-bone fractures in the pediatric age group with an incidence of 15%. These fractures are also the second most common fractures requiring hospitalization after femoral fractures. A closed reduction followed by casting is the mainstay treatment in these fractures. Though surgery is not required in most patients, it may be required in patients with open fractures, polytrauma, neurovascular injury, and unstable fractures causing unacceptable angulation. Surgical treatment can be performed with different fixation methods such as intramedullary fixation, crossed Kirschner wires (K-wires), or external fixators. External fixators are the first-line treatment in fractures with severe soft-tissue loss, although they have been associated with several complications such as delayed union, malunion, high incidence of pin tract infections, and leg-length inequality [1,2,3,4,5,6,7].

On the other hand, although rigid intramedullary nails are a popular treatment option in adults, they are not recommended in children due to an increased risk of proximal tibial epiphyseal injury. Therefore, in children, titanium elastic nails (TENs) are used instead, which are inserted into the metaphysis without causing epiphyseal injury. TENs are known to provide several advantages such as reduced soft-tissue injury, minimal proximal tibial epiphyseal injury, stable and flexible mobility, application without opening the fracture line, reduced treatment costs, and shorter operative times. Due to these advantages, TENs have become highly popular in the treatment of tibial shaft fractures [3,4,5,8,9]. Nevertheless, TENs have also been shown to have several disadvantages such as an inability to achieve stability in patients with complex fractures or severe soft-tissue injury, delayed union in advanced-age pediatric patients, and increased risk of complications such as delayed union and compartment syndrome in the treatment of complex fractures in patients older than 14 years with a body weight of over 50 kg. For these reasons, elastic nails should be utilized only in selected patients [10,11].

In this study, we aimed to compare the radiological and functional outcomes in pediatric patients with unstable closed or open fractures who could not be treated with conservative methods and underwent a fixation with TENs.

## 2. Materials and Methods

### 2.1. Research Population

The study was initiated after obtaining an approval from the local ethics committee (Approval No. 2019/2-24). The study retrospectively reviewed the medical records of 36 patients younger than 15 years who were operatively treated with TENs due to tibial shaft fractures between 2013 and 2017. Of these, 16 patients with a minimum follow-up period of 6 months, multiple fractures, head trauma, and incomplete patient records were excluded from the study. As a result, a total of 20 patients were included in the study. Open fractures were classified using the Gustilo–Anderson open fracture classification [12]. Demographic and clinical characteristics including age at injury, gender, side of the fracture, mechanism of injury, postoperative sagittal and coronal angulation, postoperative and follow-up complications, time to partial weight bearing, time to union (the appearance of bridging callus on three cortices), final coronal and sagittal angulation, knee and ankle range of motion, and leg-length discrepancies were reviewed for each patient based on the medical and radiographic records of the patients. The clinical outcomes were evaluated based on the criteria defined by Flynn et al. [13].

### 2.2. Surgical Technique

All the surgical procedures were performed under general anesthesia. In each patient, the closed reduction technique defined by Ligier and Metaizeau [8] was administered under the guidance of C-arm fluoroscopy, with the patient lying in the supine position. Medial and lateral mini incisions were made 2 cm distal to the proximal epiphysis of the tibia. After the creation of an entry point for the insertion of a TEN using a guide, an intramedullary TEN with a diameter of 2–4 mm was percutaneously inserted from the lateral side of the fracture to the fracture line, depending on the patient’s age and bone size. After the confirmation of fracture reduction with C-arm fluoroscopy, the TEN was advanced to the distal side of the fracture. Subsequently, a second intramedullary TEN with an appropriate diameter was inserted through the proximal medial of the tibia. Following the confirmation of fracture reduction and the positioning of the TENs with C-arm fluoroscopy, the tip of the TENs was bent and cut above the skin and a long leg cast/splint was applied (Figure 1).

### 2.3. Statistical Analysis

The data were evaluated using SPSS 21.0 for Windows (IBM, Armonk, NY, USA). Nonparametric data were compared using the Mann–Whitney U test and Kruskal–Wallis test. The numerical variability were expressed as mean ± standard deviation (SD). A *p* value of <0.05 was considered significant.

## 3. Results

The 20 patients comprised 16 (80%) boys and 4 (20%) girls with a mean age at injury of 8.9 ± 2.78 (range of 3–14) years. The mechanism of injury was motor vehicle accident in 12 (60%) patients and fall from heights in 8 (40%) patients. Seven (35%) out of 20 fractures were open fractures, of which 1 fracture was classified as Grade I and 6 fractures were classified as Grade II based on the Gustilo–Anderson open fracture classification. The fractures were located in the left tibia in 14 (70%) patients and in the right tibia in 6 (30%) patients. Five (25%) patients had a tibial shaft fracture alone, while 15 (75%) patients had a tibial shaft fracture accompanied by a fibular fracture (Table 1). The mean follow-up period was 24.9 ± 15.7 (range of 7–60) months. No patient had an angulation of over 10°, and 9 (45%) patients had an angulation of 5–10°. One patient had a leg-length inequality of 10 mm. Among 3 patients with open fractures, 2 of them had superficial wound infections and the other patient had a deep wound infection. Four other patients had signs of pin tract irritation that required no intervention. All the infections were successfully treated with appropriate antibiotic therapies. The mean time to partial weight bearing was 2.1 ± 0.85 (range of 1–3) weeks, and the mean time to union was 10.85 ± 3.39 (range of 6–20) weeks. No patient had a restricted range of motion of the ankle and knee joints. According to Flynn’s criteria, 12 (60%) patients had an excellent outcome and 8 (40%) patients had a satisfactory outcome (Table 2). No significant difference was observed between patients with open and closed fractures with regard to the clinical and radiological findings, although patients with open fractures had a significantly higher complication rate compared to patients with closed fractures (*p* < 0.05) (Table 3).

## 4. Discussion

A closed reduction followed by casting is the mainstay treatment in most pediatric patients with tibial shaft fractures. Surgical treatment may be required in patients with open fractures, instability, secondary reduction loss, neurovascular injury, and polytrauma. Surgical treatment can be performed with different fixation methods such as external fixators, intramedullary nails, plate/screw osteosynthesis, and percutaneous pinning [9].

Titanium elastic nails (TENs) have recently emerged as a popular treatment option in select open fractures due to several advantages including applicability in open fractures, minimal scar formation, and providing excellent outcomes in pediatric tibial shaft fractures [14,15].

Debnath et al. [16] evaluated 30 patients with midshaft tibial fractures and revealed that they obtained an excellent outcome in 50%, an acceptable outcome in 36%, and a poor outcome in 14% of the patients. Pennock et al. [17] compared patients undergoing plate/screw osteosynthesis with those undergoing TENs and reported that 97% of them had similar recovery rates. Brien et al. [18] evaluated 16 patients and indicated that all the fractures fully healed without pain over a five-year period, with no report of nonunion, limitation of movement, or refractures. Sankar et al. [19] evaluated 19 patients with unstable fractures and reported an excellent outcome in 63%, an acceptable outcome in 32%, and a poor outcome in 5% of the patients. In our study, a complete union was achieved in all the fractures with no restriction of motion or pain, an excellent outcome was obtained in 60% of patients, and an acceptable outcome was obtained in 40% of the patients.

An intramedullary fixation with TENs was initially performed in closed fractures alone. However, over time, it became preferable in open fractures as well, and similar outcomes have been reported thus far [4]. Heo et al. [5] utilized TENs in 81% of patients with an open fracture and reported that they obtained excellent outcomes in 88% of patients and satisfactory outcomes in 12% of the patients. Başaran et al. [20] evaluated the use of TENs in the treatment of patients with open tibial shaft fractures caused by high-energy motor vehicle accidents and reported that they achieved excellent outcomes in 45% of patients and satisfactory outcomes in 55% of the patients. In our study, open fractures were present in 35% of the patients and a complete union was achieved in all the fractures, whereby an excellent outcome was obtained in 43% and a satisfactory outcome was obtained in 57% of the fractures. Moreover, 15% of these fractures were classified as Grade I and 85% of them were classified as Grade II fractures. However, there were no Grade III fractures in our patients which are commonly treated with external fixators in lieu of TENs. However, although TENs have been shown to provide successful outcomes in the treatment of Grade III open fractures, which can be highly complex, we consider that TENs should be utilized only in select cases of Grade III open fractures [5]. It is commonly known that the time to union can be longer in open fractures compared to that of closed fractures. Brien et al. [18] reported that they achieved a union in an average of 9 weeks in closed fractures and in an average of 15 weeks in open fractures. In our patients, the mean time to union was longer in open fractures compared to closed fractures (11.28 ± 2.21 vs. 10.61 ± 3.94 weeks), although no significant difference was found. Meaningfully, patients with open fractures should be monitored more carefully due to the higher risk of a delayed union.

In patients with tibial shaft fractures, the time to union may vary depending on age and is likely to be prolonged particularly in advanced ages. Gordon et al. [10] evaluated 50 tibial fractures and reported that the mean age was 11.7 years in patients with a normal time to union as opposed to 14.1 years in patients with a delayed union. In our study, the mean time to union was 10.46 ± 4.09 weeks in patients younger than 10 years as opposed to 11.57 ± 1.39 weeks in patients older than 10 years. Based on these findings, it is recommendable that a delayed union should be kept in mind in advanced-age children treated with TENs.

Titanium elastic nails (TENs) have a higher possibility of causing malunion by allowing angulation during union compared to plate-screw fixation and rigid fixations. Brien [18] et al. utilized TENs and reported an angulation of over 5° in 2 (12.5%) of the patients, with one patient having an angulation of 6° in the coronal plane and the other patient having an angulation of 10° in the sagittal plane. Sankar et al. [19] reported that 1 (6.3%) patient had an angulation of 5–10° in the sagittal plane and that 3 (18.9%) patients had an angulation of 5–10° in the coronal plane. In our patients, no patient had an angulation of over 10° in the final examination, while 9 (45%) patients had an angulation of 5–10°. The higher angulations in our patients could be related to the follow-up periods of the patients and to the fact that children have a higher remodeling capacity compared to adults. The mean follow-up periods were 31 months in patients with no angulation and 22.5 months in patients with an angulation. Moreover, the mean follow-up period was less than 20 weeks in 6 out of 9 patients with a final angulation.

A treatment with TENs is likely to cause pin tract infection or irritation. Onta et al. [21] and Sankar et al. [19] reported that 22% and 26% of patients had pin tract irritation, respectively. Mani et al. [22] reported that 13.3% of patients had pin tract irritation and 4.4% had superficial pin tract infection. In our study, pin tract irritation developed in 20% of the patients and healed spontaneously without causing any problems after the removal of the TENs. Additionally, superficial pin tract infections were observed in 2 (10%) of our patients with open fractures and these patients were treated with antibiotherapy with no need for any additional interventions. In our clinic, we usually leave the TEN ends on top of the skin due to the limited soft-tissue support in the proximal tibia, which could explain the occurrence of the pin tract infections in two of our patients and in no other patients in similar studies [5,10,17]. However, the original method described by Ligier et al. [8] proposes that the TEN ends should be placed under the skin since the TEN ends left above the skin increase the risk of infections. Therefore, care should be taken to leave the TEN ends under the skin.

Leg-length inequality is among the complications that can be experienced after tibial shaft fractures. Walamastha et al. [23] reported that they found a leg-length inequality of <15 mm in 3.6% of the patients. In our study, only one (5%) patient with a Grade II fracture was detected with an asymptomatic leg-length inequality of 10 mm.

## 5. Conclusions

In conclusion, an intramedullary fixation with TENs provides favorable outcomes in the treatment of unstable pediatric tibial shaft fractures that cannot be reduced with conservative treatment modalities or cannot be casted due to the presence of an edema or open wound.

## Figures and Tables

**Figure 1 medicina-55-00266-f001:**
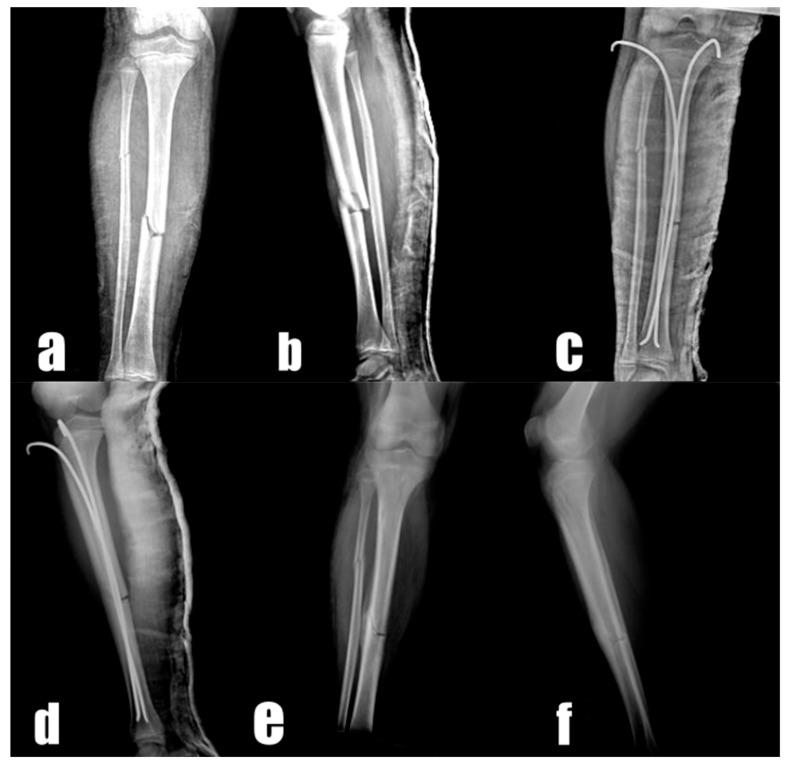
Radiographs of a 11-year-old boy: preoperative anteroposterior (**a**) and lateral (**b**), postoperative first-day anteroposterior (**c**) and lateral (**d**), and postoperative 8th-week anteroposterior (**e**) and lateral (**f**) radiographs.

**Table 1 medicina-55-00266-t001:** Demographic characteristics of patients.

Patients	Gender	Age (year)	Side	Character of Fracture	Fracture Type	Etiology	Fracture
1	M	11	R	Closed	Transverse	Falling Down	Tibia-Fibula
2	M	13	L	Closed	Oblique	Road Accident	Tibia-Fibula
3	M	14	L	Closed	Transverse	Road Accident	Tibia-Fibula
4	F	3	R	Closed	Oblique	Road Accident	Tibia-Fibula
5	F	12	L	Closed	Spiral	Falling Down	Tibia
6	M	10	L	Closed	Spiral	Road Accident	Tibia-Fibula
7	M	8	L	Closed	Spiral	Falling Down	Tibia-Fibula
8	M	8	L	Closed	Spiral	Falling Down	Tibia
7	F	9	L	Closed	Spiral	Falling Down	Tibia
10	F	7	L	Closed	Oblique	Road Accident	Tibia-Fibula
11	M	5	L	Closed	Transverse	Road Accident	Tibia-Fibula
12	M	5	L	Closed	Spiral	Road Accident	Tibia-Fibula
13	M	9	R	Closed	Spiral	Falling Down	Tibia
14	M	7	L	GRADE 1	Transverse	Falling Down	Tibia-Fibula
15	M	8	L	GRADE 2	Spiral	Road Accident	Tibia-Fibula
16	M	11	R	GRADE 2	Transverse	Road Accident	Tibia-Fibula
17	M	12	R	GRADE 2	Transverse	Road Accident	Tibia-Fibula
18	M	8	L	GRADE 2	Spiral	Road Accident	Tibia
19	M	9	L	GRADE 2	Oblique	Falling Down	Tibia-Fibula
20	M	9	R	GRADE 2	Spiral	Road Accident	Tibia-Fibula

M: male, F: female, R: right, L: left.

**Table 2 medicina-55-00266-t002:** A comparison of the radiological and clinical results of the patients.

Patients	Character of Fracture	Follow up (Month)	Healing Time	Residual Deficiency	Complication	Weight-Bearing Time (Week)	Pin Tract Infection/Irritation	Shortness	Functional Result (Flynn Criteria)
1	Closed	22	10	0/0	None	2	Irritation	None	Excellent
2	Closed	60	11	0/0	None	1	None	None	Excellent
3	Closed	30	12	0/0	None	2	None	None	Excellent
4	Closed	30	10	6/3	None	3	None	None	Satisfactory
5	Closed	7	14	6/8	None	2	None	None	Satisfactory
6	Closed	16	12	9/7	None	2	None	None	Satisfactory
7	Closed	18	14	4/8	None	2	Irritation	None	Satisfactory
8	Closed	42	7	0/8	None	2	None	None	Excellent
7	Closed	40	6	0/0	None	1	None	None	Excellent
10	Closed	22	20	0/0	None	1	None	None	Excellent
11	Closed	7	7	4/4	None	3	Irritation	None	Excellent
12	Closed	9	6	0/0	None	1	None	None	Excellent
13	Closed	9	9	3/5	None	2	None	None	Excellent
14	Grade I Open	28	8	0/0	None	1	None	None	Excellent
15	Grade II Open	40	10	0/0	None	2	None	None	Excellent
16	Grade II Open	15	10	9/3	None	3	Irritation	None	Satisfactory
17	Grade II Open	20	12	6/3	Wound infection (Superficial)	2	None	None	Satisfactory
18	Grade II Open	48	12	8/0	None	3	None	None	Satisfactory
19	Grade II Open	7	15	10/5	Wound infection (Deep)	4	infection	Yes	Satisfactory
20	Grade II Open	28	12	0/0	Wound infection (Superficial)	3	infection	None	Excellent

**Table 3 medicina-55-00266-t003:** A comparison of patients with closed and open fractures.

Features	Closed Fracture	Open Fracture	*p* Value
Patients	13	7	
Gender M/F	9/4	7/0	*p* > 0.05
Side R/L	3/10	3/4	*p* > 0.05
Weight-bearing time (week)	1.84 ± 0.68	2.57 ± 0.97	*p* > 0.05
Healing time (week)	10.61 ± 3.93	11.28 ± 2.21	*p* > 0.05
Complication	None	3	*p* < 0.05
Residual deficiency (coronal/sagittal angulation) degree	5	4	*p* > 0.05
Pin tract infection/irritation	3	3	*p* > 0.05
Shortness	No	1	*p* > 0.05
Functional result(Flynn Criteria)	Excellent: 9	Excellent: 3	*p* > 0.05
	Satisfactory: 4	Satisfactory: 4	

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
