# Peer review of "Treatment of Unstable Pediatric Tibial Shaft Fractures with Titanium Elastic Nails"

_medicina, 2019, doi:10.3390/medicina55060266_

Round 1

Reviewer 1 Report

The article is a case series about tibial shaft fractures in pediatric age treated with elastic titanium nail.

Introduction is too short. The authors need to provide more information about the topic (etiology, peak age incidence, treatment options, literature evidence about titanium elastic nail).

Methods are poor. I think they need a logical order of appearance. No statistical analysis is described.

Results and Data in general should be more scientific: quantitative data must be described as mean, standard deviation and range. 

Statistical analysis is necessary to validate a study.

In conclusion the sentence"reductions is a good surgical treatment method in cases with open fractures where conservative treatment is unsuccessful" is not exact. 

For open fracture the first choice is external fixation. Successively, you can use the elastic nail.

More recent references could be considered.

Author Response

Dear Reviewer, I would initially like to present my thanks for your valuable and contributory remarks. In line with your remarks, the following improvements and/or additions were performed in the manuscript:

1. The Introduction section was revised as per your remarks. In the same section, treatment options were exemplified and it was also noted that although elastic nails can be used in patients with unclosed epiphysis, there is an increased risk of complications such as delayed union and compartment syndrome in the treatment of complex fractures particularly in patients older than 14 years with a body weight of over 50 kg.

2. The treatment methods were elaborated and improvements were performed with the addition of further notions. The statistical analysis was separated into a subsection under the Materials and Methods section. Findings were presented systematically and divided three tables

3. Numerical findings were expressed with standard deviation and range.

4. In the Conclusion section, the term “a good treatment method” was changed to “a favorable treatment method”.

5. The overall perspective of the study was modified from “a general evaluation of the patients” to “a statistical comparison of open and closed fractures” and this modification was reflected in both Findings and Discussion sections.

6. It was clearly declared that external fixators are the first-line treatment in complex open fractures and are commonly preferred by surgeons. This notion was exemplified by noting that there were no Grade III fractures in our patients.

7. The references section was further enriched.

8. The entire manuscript was revised with regard to language and grammar.

Thanks in advance for your further consideration,

Yours sincerely.

Reviewer 2 Report

The authors present a single center, single arm, retrospective, observational study about the efficacy of ESIN Treatment in school-Age children.

Although the study has been conducted well I am sorry to conclude that such data have  been published many times since  Ligier introduced ESIN in 1982 (Chir Pediatr; 24:383-5.

- Sim E. Arch Orthop Trauma Surg 1991

This encludes reviews like Schmidt AH. Instr Course Lect 2003;52:607-22

This includes long-term observations such as Vallamshetla VR. J Bone Joint Surg Br 2006;88:536-40

Even the outcome of ESIN with/Without additional fracture of the fibula have been explored (Canavese F et al. J Pediatr Orthop 2016).

So I believe only a substatial modification of the study design could improve the scientif value of this manuscript.

Author Response

Dear Reviewer, I would initially like to present my thanks for your valuable and contributory remarks. In line with your remarks, the following improvements and/or additions were performed in the manuscript:

1. The overall perspective of the study was modified from “a general evaluation of the patients” to “a statistical comparison of open and closed fractures” and this modification was reflected in both Findings and Discussion sections.

2. The treatment methods were elaborated and improvements were performed with the addition of further notions. The statistical analysis was separated into a subsection under the Materials and Methods section. Numerical findings were expressed with standard deviation and range. Findings were presented systematically and divided three tables

3. The entire manuscript was revised with regard to language and grammar.

Thanks in advance for your further consideration,

Yours sincerely.

Reviewer 3 Report

What are the indications for fixing these fractures.

The xray depicted show acceptable position

Why was this fracture fixed.

Author Response

Dear Reviewer, many thanks indeed for praising our study.

1.The images given in the study pertain to a Grade II patient with dermabrasion and edema. We had to perform surgery in this patient due to the presence of edema, skin disorders, and a previously sutured open wound.

2. The entire manuscript was revised with regard to language and grammar.

Thanks in advance for your further consideration,

Yours sincerely.

Round 2

Reviewer 1 Report

The text has been improved. I think it could be suitable for publication, but with an improvement of English language. 

Reviewer 2 Report

The authors present a single-center, one-armed, retrospective, observational study about the complication rate of ESIN in pediatric unstable tibial shaft fractures.

Certainly this topic is not new (at least 40 hits in PuBMed) and the study design cannot not deliver high-lever of evidence.

Nevertheless the manuscript has been prepared well and maybe published in its present form.